# microRNA Expression Profile in Obesity-Induced Kidney Disease Driven by High-Fat Diet in Mice

**DOI:** 10.3390/nu16050691

**Published:** 2024-02-28

**Authors:** Àuria Eritja, Maite Caus, Thalia Belmonte, David de Gonzalo-Calvo, Alicia García-Carrasco, Ana Martinez, Montserrat Martínez, Milica Bozic

**Affiliations:** 1Vascular and Renal Translational Research Group, Biomedical Research Institute of Lleida Dr. Pifarré Foundation (IRBLleida), 25196 Lleida, Spain; aeritja@irblleida.cat (À.E.); mcaus@irblleida.cat (M.C.); agarcia@irblleida.cat (A.G.-C.); anamaria.martinez@udl.cat (A.M.); 2Translational Research in Respiratory Medicine, Hospital Universitari Arnau de Vilanova-Santa Maria, Biomedical Research Institute of Lleida (IRBLleida), 25198 Lleida, Spain; thaliabelmonte@gmail.com (T.B.); dgonzalo@irblleida.cat (D.d.G.-C.); 3CIBER of Respiratory Diseases (CIBERES), Institute of Health Carlos III, 28029 Madrid, Spain; 4Biostatistics Unit (Biostat), Biomedical Research Institute of Lleida Dr. Pifarré Foundation (IRBLleida), 25196 Lleida, Spain; mmartinez@irblleida.cat

**Keywords:** kidney, obesity, chronic kidney disease, high-fat diet, lipotoxicity, miRNA, obesity-induced kidney disease, next-generation sequencing, miRNA-seq

## Abstract

Obesity is one of the main causes of chronic kidney disease; however, the precise molecular mechanisms leading to the onset of kidney injury and dysfunction in obesity-associated nephropathy remain unclear. The present study aimed to unveil the kidney microRNA (miRNA) expression profile in a model of obesity-induced kidney disease in C57BL/6J mice using next-generation sequencing (NGS) analysis. High-fat diet (HFD)-induced obesity led to notable structural alterations in tubular and glomerular regions of the kidney, increased renal expression of proinflammatory and profibrotic genes, as well as an elevated renal expression of genes involved in cellular lipid metabolism. The miRNA sequencing analysis identified a set of nine miRNAs differentially expressed in the kidney upon HFD feeding, with miR-5099, miR-551b-3p, miR-223-3p, miR-146a-3p and miR-21a-3p showing the most significant differential expression between standard diet (STD) and HFD mice. A validation analysis showed that the expression levels of miR-5099, miR-551b-3p and miR-146a-3p were consistent with NGS results, while Kyoto Encyclopedia of Genes and Genomes (KEGG) and Gene Ontology (GO) enrichment analyses revealed that these three validated miRNAs modulated target genes involved in metabolic and adipocytokine pathways, fatty acid and lipid metabolism, and inflammatory, senescence and profibrotic pathways. Our results suggest that differentially expressed miRNAs play pivotal roles in the intricate pathophysiology of obesity-associated kidney disease and could potentially create novel treatment strategies to counteract the deleterious effects of obesity on kidney function.

## 1. Introduction

Obesity is an escalating worldwide health concern which poses a significant challenge to global well-being. It is recognized as a persistent and relapsing condition [1], as well as a risk factor for a myriad of other chronic noncommunicable diseases, with elevated rates of disability and overall mortality [2,3]. In recent decades, there has been a notable rise in the prevalence of obesity-associated complications, while obesity has been recognized as an independent risk factor for the development of chronic kidney disease (CKD) [4,5]. Indeed, the occurrence of obesity-associated kidney disease has risen significantly during the past few years, showing a 10-fold increase [6]. Obesity-associated kidney disease is characterized by tubular and glomerular hypertrophy, lipid accumulation in the renal parenchyma, mesangial expansion, glomerular hyperfiltration and alterations of renal vasculature that, alongside renal inflammation, oxidative stress and impaired mitochondrial homeostasis [7], collectively contribute to kidney dysfunction and the development of CKD [8]. Recent studies have suggested that disruptions in the metabolism of fatty acids and cholesterol play an important role in the abnormal build-up of lipids in the kidney and lipotoxicity [7,9]. Indeed, excess fatty acids in tubular cells which surpass the β-oxidative capacity of the mitochondria result in lipid deposition within these cells [7]. Excessive accumulation of lipids in cells that lack an appropriate mechanism to tackle immense lipid loads [10,11], such as native kidney cells [12], may prompt a series of events encompassing changes in various cell signaling pathways, the release of proinflammatory and profibrotic molecules [13], as well as the production of reactive oxygen species [14], which over time contribute to the development of glomerulosclerosis and tubulointerstitial fibrosis (TIF), thus worsening the kidney damage.

In spite of the multitude of studies linking obesity and elevated lipid levels to kidney damage, the exact molecular mechanism responsible for the development of kidney dysfunction remains incompletely understood. Thus, it is imperative to explore novel potential treatment avenues focused on preventing or counteracting the harmful impacts of obesity and irregular lipid levels on kidney function.

MicroRNAs (miRNAs) are small, non-coding, single-stranded RNA with the crucial role in post-transcriptional regulation of gene expression [15]. Furthermore, their role in transcriptional gene activation or silencing has also been reported [16]. miRNAs play a crucial role in normal development and preserving homeostasis, actively engaging in basic cellular functions like proliferation, differentiation, apoptosis and metabolism [17,18]. Of note, miRNAs can serve as diagnostic and prognostic markers in a range of diseases such as cardiovascular disorders, kidney disease, cancer and autoimmune diseases [19,20,21].

miRNAs have been implicated in the development and progression of obesity, and several candidates have been identified as critical regulators of adipogenesis and lipid metabolism [22,23], while their dysregulation contributes to obesity-related metabolic abnormalities [24]. Altered expression of miRNAs in obesity can exert a direct impact on kidney function by promoting essential features of CKD such as inflammation, oxidative stress and TIF [25,26]. Of interest, miRNAs have been demonstrated to be important players in the onset and development of various renal conditions such as renal TIF, acute kidney injury, diabetic nephropathy (DN), lupus nephritis, IgA nephropathy, polycystic kidney disease, etc. [18,19,27,28]. A variety of miRNAs such as the miR-29a family [19,29,30] and miR-30 family [31], miR-21 [32], miR-199, miR-200, miR-214, miR-382, miR-133, miR-212 [31], mir-433 and miR-192 [33], and miR-184 [19] have been shown to be involved in the process of tubular epithelial–mesenchymal transition (EMT) and kidney TIF. Moreover, their role in the development of DN has also been extensively investigated, with the miR-21, miR-25 and miR-29 families, miR-34a-5p, miR-141, miR-146a, miR-184, miR-370 and miR-377, among others, having pivotal roles, as reviewed previously [19]. However, the knowledge of miRNAs’ expression pattern and their functions in obesity-related kidney disease is still limited. Unraveling the miRNA expression profile in obesity and CKD can enhance our comprehension of the regulatory functions of miRNAs in these interconnected conditions, offering the potential for more precise diagnosis and targeted therapeutic interventions.

In order to enhance our understanding of the pathophysiological pathways underlying the initiation and progression of obesity-associated nephropathy, the aim of this study was to analyze by next-generation sequencing (NGS) the renal miRNA expression profile in a mouse model of obesity-induced kidney disease (OIKD) driven by high-fat diet (HFD) feeding.

## 2. Results

### 2.1. Metabolic and Physiological Parameters Induced by High-Fat Diet

After 10 weeks of an HFD, mice exhibited significant hypercholesterolemia, characterized by elevated levels of total cholesterol and low-density lipoprotein (LDL) and high-density lipoprotein (HDL) cholesterol, as well as triglycerides significantly higher than in a standard diet (STD)-fed group of mice (Table 1). Prominent dyslipidemia in obese mice was concomitant with body weight gain that was 27.4% greater (*p* < 0.0001) than that in the STD group, despite a slight decrease in food intake (Table 1). HFD feeding induced an increase in fasting glycemia (*p* = 0.0229) compared with the STD group (Table 1), while blood urea nitrogen (BUN) did not show significant changes at the end of the HFD feeding (Table 1). HFD-fed mice showed an over 2-fold increase in serum C-peptide (*p* = 0.0288) and over 3-fold increase in HOMA-IR C-peptide index (*p* = 0.0206), compared to the STD group (Table 1). In addition, HFD feeding led to a 2-fold increase in serum AST levels (*p* = 0.0014), whereas the increase in ALT levels was not statistically significant (*p* = 0.0508) (Table 1).

### 2.2. Kidney Injury and Dysfunction in OIKD Model

In our OIKD mouse model, HFD feeding led to a significant increase in neutral lipids accumulation in the tubular area of the kidney (5.31 ± 0.782; *p* = 0.0079) (Figure 1A–C), as well as clear morphological changes and damage of renal parenchyma compared with the STD group (Figure 1D). Histopathological analysis of renal tissue revealed that mice fed an HFD developed larger glomeruli (Figure 1G,J), alongside the mesangial matrix expansion (MME) and irregular thickening of the glomerular basement membrane (GBM) (Figure 1E). Furthermore, HFD induced an increase in cytoplasmic vacuolation of renal tubular cells localized in the kidney cortex, specifically proximal tubular cells (Figure 1H,K). As depicted in Figure 1H,K, numerous proximal tubular cells seemed to be entirely occupied by vacuoles, leading in some instances to a loss of brush border. Next, we analyzed expression levels of adiposity-related proinflammatory and profibrotic genes in the kidney. HFD induced a significant increase in renal MCP1 (9.97-fold; *p* < 0.0001), RANTES (1.97-fold; *p* = 0.0001), TNFα (1.93-fold; *p* = 0.0003) and iNOS (1.49-fold; *p* = 0.002) mRNA in HFD-fed mice, compared with their STD littermates (Figure 2A–C,G). Moreover, obese mice fed an HFD exhibited elevated expression of renal αSMA (1.27 ± 0.099; *p* = 0.0479), fibronectin (1.62 ± 0.109; *p* = 0.0011) and TGFβ mRNA (1.51 ± 0.106; *p* = 0.0033) (Figure 2D–F), whereas no differences were detected in the mRNA levels of collagen I α1 or collagen I α2 in the kidney.

Next, we analyzed the expression levels of two obesity-related genes, FATP2 and SREBP1, known to be involved in the regulation of cellular lipid metabolism and kidney homeostasis [34,35], as well as in the pathogenesis of CKD [34,36]. Indeed, HFD-fed mice showed an increase in the expression of fatty acid transporter FATP2 mRNA (1.31 ± 0.037; *p* = 0.0007) (Figure 2H) and lipogenesis regulator SREBP1 mRNA (1.21 ± 0.0618; *p* = 0.0345) (Figure 2I) in the kidney, compared to the STD group.

### 2.3. Differential microRNA Expression Profile in Kidneys of OIKD Model

To assess miRNAs implicated in the pathogenesis of OIKD driven by HFD in C57BL/6J mice, we performed miRNA-seq study to detect miRNAs in the kidney tissue of STD- and HFD-fed mice. After miRNA expression profiling, unsupervised hierarchical cluster analysis of the 35 most variable miRNAs revealed 19 miRNAs whose expression was downregulated and 16 miRNAs with an upregulated expression in the kidney upon HFD feeding (Figure 3). Differential gene expression analysis identified 9 significantly differentially expressed miRNAs (FDR *p* < 0.05) (Table 2). To validate miRNA-seq results, we selected 5 miRNAs that showed the most significant differential expression between STD and HFD mice, namely, miR-5099, miR-551b-3p, miR-223-3p, miR-146a-3p and miR-21a-3p (FDR *p* < 0.01) (Table 2), and their expression levels were assessed by qPCR. Furthermore, to accurately quantify miRNA expression levels, we first performed the analysis of stably expressed miRNAs in order to find the most appropriate endogenous control for our OIKD model. We used data from miRNA-seq analysis and evaluated the expression stability of endogenous control candidates using the RefFinder software (https://github.com/fulxie/RefFinder, accessed on 1 November 2023). We identified miR-30a-3p and miR-30b-5p among the top five stable candidates in our dataset and we confirmed their expression by qPCR.

The qPCR validation of miRNA-seq analysis confirmed that the expression levels of three miRNAs, miR-5099, miR-551b-3p and miR-146a-3p, were consistent with the NGS results (Figure 4A–C). However, expression levels of miR-21a-3p and miR-223-3p were not confirmed by qPCR analysis (Appendix A). Thus, miR-551b-3p and miR-5099 were significantly downregulated, whereas the miR-146a-3p was significantly upregulated, in kidneys of HFD-fed mice compared with their STD littermates. Importantly, miR-5099, miR-551b-3p and miR-146a-3p expressions in the kidney showed consistency after normalization with both endogenous controls, miR-30a-3p and miR-30b-5p (Figure 4A–C).

### 2.4. Correlation between miRNA Expression, Total Serum Cholesterol and Renal Lipid Content

The correlation analysis showed an inverse association between the miR-5099 expression and total serum cholesterol levels (R^2^ = 0.46; R = −0.6811; *p* < 0.003), as well as the renal lipid content measured by Oil red O staining (R^2^ = 0.50; R = −0.707; *p* < 0.003) (Appendix A, respectively). Furthermore, we found a significant association between miR-146a-3p expression levels and total serum cholesterol levels (R^2^ = 0.352; R = 0.5930; *p* < 0.0155), but not with the renal lipid content (R^2^ = 0.143; R = 0.3776; *p* < 0.149) (Appendix A, respectively). However, we did not detect any association between the miR-551b-3p expression and the total serum cholesterol (R^2^ = 0.007; R = −0.086; *p* = 0.743), nor with renal lipid content (R^2^ = 0.05125; R = −0.2264; *p* = 0.3823) (Appendix A, respectively).

### 2.5. Functional Enrichment Analyses

In order to elucidate how differentially expressed miRNAs in our mouse model might contribute to the inflammatory and profibrotic responses in the obesity-induced kidney injury, we searched for target genes of three validated miRNAs, miR-5099, miR-551b-3p and miR-146a-3p, using the online computational resource DIANA-miRPath v4.0 [37]. This tool integrates data on predicted miRNA targets from TargetScan v8.0, along with functional enrichment analyses based on the Kyoto Encyclopedia of Genes and Genomes (KEGG) and Gene Ontology (GO) databases. A total of 6412 target genes were identified for the three validated miRNAs. The KEGG pathway analysis showed that 124 pathways were enriched in putative target genes of our validated miRNA candidates (Appendix A). Differentially expressed miRNAs were potentially involved in pathways such as Hippo signaling pathway, chemokine signaling pathway, metabolic pathways, Ras and MAPK signaling pathways, FoxO signaling pathway, adipocytokine signaling pathway, and fatty acid metabolism, PI3K-Akt, NFκB and TGFβ signaling pathways (Figure 5A). 

The Gene Ontology (GO) analysis revealed that the target genes were significantly enriched in 482 biological processes (Appendix A). The most relevant biological processes between the STD and HFD experimental groups were cytoplasm, lipid metabolic process, fatty acid metabolic process, lipid binding, cellular senescence, extracellular matrix organization, TGFβ receptor signaling process, positive regulation of MAPK cascade, SMAD protein signal transduction, etc. (Figure 5B).

Next, we assessed the activities of the MAPK, PI3K-Akt and NFκB pathways, which had been identified through our bioinformatic analysis and are recognized for their critical roles in the initiation and progression of kidney disease [38]. We found a marked decrease in κB inhibitor (IκB)α protein, a key molecular target regulating NFκB activation [39], and a significant increase in the NFκB p50 subunit in the kidneys of mice fed an HFD, compared with the STD group (Figure 6A–C).

Moreover, HFD feeding resulted in an elevation in pAkt, as seen by Western blot analysis (Figure 6A); however, it did not exert any influence on the activity of ERK1/2 and p38 (Figure 6A).

## 3. Discussion

Obesity stands as one of the potent contributors to CKD; however, the exact pathophysiological pathways underlying the onset and progression of obesity-related nephropathy remain elusive. In this study, we established a mouse model of OIKD by feeding C57BL/6J mice with a high-fat diet. The C57BL/6J strain stands out as highly prone to obesity and OIKD when fed an HFD or Western-style dietary regimen, displaying early signs of renal pathophysiological changes associated with obesity, as well as marked proinflammatory and profibrotic microenvironment [40,41,42,43,44]. Being characterized by systemic changes and kidney damage that closely mimic those observed in human patients with this condition, a mouse model of OIKD is a valuable tool for conducting preclinical research related to obesity and its associated renal complications [40,41,42,43]. In our model, HFD feeding resulted in an increase in body weight, hypercholesterolemia and hyperglycemia. Additionally, HFD induced increased serum C-peptide levels and elevated HOMA-IR index, indicating insulin resistance in our model. This was accompanied by structural changes in the kidneys and an upregulation of proinflammatory, profibrotic and obesity-related genes in the renal tissue. 

In the current study, we conducted miRNA-seq analysis and revealed a set of nine miRNAs significantly differentially expressed in kidneys of mice fed an HFD compared to control littermates fed an STD. The qPCR validation confirmed miRNA-seq results, demonstrating a significant decrease in miR-5099 and miR-551b-3p, and an increase in miR-146a-3p, in kidneys of HFD-fed mice. Of note, miRNA expression in our mouse model showed consistency after normalization with two different endogenous controls. The selection of proper endogenous control candidates is essential for qPCR studies, as housekeeping genes can exhibit variability and are susceptible to directional shifts induced by experimental conditions [45]. The use of more than one endogenous control improves the reliability and precision of quantitation compared to the use of a single endogenous control [46]. In this study, we identified two endogenous control genes, miR-30a-3p and miR-30b-5p, for qPCR analysis of miRNA expression in the OIKD mouse model that can be used in a wide array of future research applications in the field. Furthermore, bioinformatic analysis pointed to inflammatory and metabolic pathways as the most relevant pathways associated with the validated miRNA candidates, emphasizing the importance of inflammation and lipid metabolism in the pathogenesis of obesity-related nephropathies.

Among the miRNAs that exhibited differential expression in our study, miR-5099 was significantly decreased in kidneys upon HFD feeding. There are limited data on the role of miR-5099 in different tissues. Namely, low levels of miR-5099 have been described in joint tissue of old mice [47], while high levels of miR-5099 have been demonstrated in ischemic stroke [48], liver injury [49] and podocyte injury induced by puromycin aminonucleoside (PAN) [50]. Our results point to the potential participation of miR-5099 in the onset and development of OIKD, as its expression was not only downregulated in kidneys affected by obesity, but was also inversely associated with total serum cholesterol and renal lipid content, supporting KEGG and GO enrichment analysis which revealed the involvement of metabolic and fatty acid metabolism pathways, as well as lipid metabolic process.

Another miRNA found to be significantly downregulated in kidneys upon HFD feeding in our study was miR-551b-3p. miR-551b-3p has been reported to be involved in different types of cancers [51,52,53], and in the pathogenesis of cardiomyopathy [54]. Interestingly, circulating miR-551b-3p has been found to be downregulated in patients with childhood obesity [55], while, on the other hand, it was upregulated in adipose tissue of obese individuals [56]. In addition, brain miR-551b expression has been shown to be modulated by an HFD in mice [57]. However, its role in kidney disease has not yet been described. Our results provide novel findings on the possible role of miR-551b-3p in the pathogenesis of OIKD.

miR-146a-3p was another candidate miRNA shown to be significantly differentially expressed in our model of OIKD. miR-146a has been reported to be involved in the pathogenesis of diabetic nephropathy [8,19,58], where it was found to be reduced in glomeruli of diabetic patients and diabetic mice [58]. Conversely, miR-146a increased in human and rat kidneys affected by type 2 DN [59]. Accordingly, our results demonstrate an increased expression of miR-146a-3p in kidneys of HFD-fed mice which significantly correlated with total serum cholesterol levels. Our study provides important confirmation for the role of miR-146a-3p in the pathogenesis of OIKD, as we used an experimental model of OIKD induced by HFD and not the one induced by streptozotocin used in the previous report [59].

Another two miRNA candidates, miR-21a-3p and miR-802-5p, both upregulated in the kidneys of HFD-fed mice in our miRNA-seq study, have also been implicated in the pathogenesis of obesity-associated renal dysfunction [26,60]. In particular, miR-21 was shown to be upregulated in kidneys of obese mice [60], while Sun et al. [26] demonstrated an increase in miR-802 expression in kidneys of mice fed an HFD, as well as in the serum of obese patients. Importantly, both research groups used an obesity model induced by HFD in mice. Of note, other candidate miRNAs have been found to been differentially expressed in our OIKD mouse model, such as miR-223-3p, miR-129-5p, miR-142a-5p and miR-144-3p. Interestingly, a meta-analysis of profiling studies in both animal models and humans has revealed the upregulation of miR-142–3p, miR-142–5p, and miR-223–3p in renal fibrosis [61]. Particularly, miR-223 has been implicated in disorders such as type II diabetes, vascular calcification and atherosclerosis [62]. Indeed, miR-223 was found to be upregulated in the calcified aortas of mice with CKD, while the serum levels of the same miRNA decreased in both mice [63] and humans with CKD [64]. Furthermore, Anglicheau et al. [65] demonstrated that the expression levels of miR-142–5p and miR-223, among others, could serve as predictors for renal graft function.

The precise mechanisms by which an HFD influences miRNA expression in the kidney are unclear, with alterations in miRNA expression potentially arising directly from the diet or indirectly from HFD-induced metabolic changes. Various studies have illustrated that obesity and an HFD can lead to kidney dysfunction by influencing diverse pathways, such as inflammation [25,26,44], stimulation of reactive oxygen species (ROS) generation [13,14] and TIF [26,43,66]. Furthermore, dietary fats, by contributing to endothelial dysfunction, are implicated in the development of hypertension, which, in turn, may serve as a secondary contributor to differential miRNA expression in the kidney. Considering these documented studies, it is plausible that alterations in multiple biological processes contribute to the effect of an HFD on renal miRNA expression observed in our study. Consequently, additional research is warranted to unravel the potential mechanisms responsible for mediating the impact of an HFD on the distinctive signature of renal miRNA expression.

Herein, we performed KEGG and GO enrichment analyses to identify possible molecular pathways and biological processes affected by dysregulated miRNA candidates in our study. KEGG pathway analysis found that three validated miRNA candidates in our study, miR-5099, miR-551b-3p and miR-146a-3p, were significantly mapped to the chemokine signaling pathway, metabolic pathways, adipocytokine signaling pathway, fatty acid metabolism and NFκB signaling pathway. Indeed, in our mouse model of OIKD, an HFD induced a significant increase in adiposity-related proinflammatory chemokines, MCP1 and RANTES, that belong to the chemokine signaling pathway [67]. Of note, TNFα and iNOS, two known downstream targets for NFκB transcription factor [68,69], were also found to be affected by HFD feeding in our study, alongside the NFκB pathway activation. Huang et al. [59] demonstrated a significant elevation in miR-146a in type 2 diabetic nephropathy, a renal condition associated with obesity and renal lipid accumulation. In their study, increased expression of miR-146a contributed to the elevated levels of TNFα and TGFβ, intensifying renal inflammation and fibrosis through the activation of the NFκB signaling pathway. Notably, our OIKD model exhibited increased levels of miR-146a-3p in the kidneys of mice exposed to an HFD, reinforcing the connection with NFκB pathway activation. This parallelism underscores the potential role of miR-146a-3p in mediating the effects of obesity-induced kidney dysfunction. Other signaling pathways enriched in our miRNA-seq study were the MAPK, PI3K-Akt and TGFβ signaling pathways. Certainly, in our study, obese mice fed an HFD showed elevated levels of renal αSMA, Fibronectin and TGFβ, known target genes of the TGFβ signaling pathway [70,71,72,73]. Furthermore, MAPK and PI3K/Akt pathways are well-known signaling pathways implicated in the pathogenesis of different nephropathies, as well as in processes pertaining to lipid metabolism [74,75]. In our study, HFD-feeding led to an increase in the activity of Akt in the kidney, while it did not have a marked effect on pERK1/2 or p38. PI3K-Akt has been shown to regulate lipid metabolism, among other genes, through SREBP1 [76], an important obesity-related gene, which was found to be upregulated in kidneys of our OIKD model. Notably, Lovis et al. [77] found that prolonged exposure of β-cell lines and pancreatic islets to saturated fatty acids resulted in an elevation in miR-146 expression. Concurrently, miR-146a-5p demonstrated the ability to modulate the PI3K-Akt [78,79] and SREBP pathways [80]. SREBPs are known to play a role in mediating lipotoxicity, thus contributing to the progression of kidney disease [35]. SREBPs are synthesized in the endoplasmic reticulum and subsequently translocated to the Golgi through a complex regulatory machinery [81], which is influenced by cellular levels of lipids and sterols, as well as factors such as the mTOR pathway [82], one of the enriched pathways in the KEGG analysis done in our miRNA-seq study. Furthermore, GO enrichment analysis pointed to cellular components such as endoplasmic reticulum and Golgi apparatus as some of the 15 top enriched cellular components, alongside other GO terms such as the lipid metabolic process, fatty acid metabolic process, lipid binding, cellular senescence, extracellular matrix organization, TGFβ receptor signaling process and SMAD protein signal transduction. There are scarce data in the literature linking miRNAs with lipid metabolic processes in the kidney. By revealing the link between miRNA expression and lipid and fatty acid metabolism in kidney tissue, our research represents a novel milestone and opens new avenues to investigate the progression of obesity-induced kidney disease.

Our study has a few potential limitations that could be addressed in future research. Firstly, the bioinformatic analysis was based on target prediction. While specific signaling pathways were explored in this study, it is essential for future research to experimentally validate additional molecular pathways and biological processes identified through computational analyses. Secondly, the demonstration of renal miRNA expression profiles in a mouse model of OIKD was conducted in vivo. To clarify the specific roles of the identified miRNAs and affirm their functional significance, it is advisable to conduct functional validation of candidate miRNAs in future in vitro studies. Addressing these limitations will contribute to the robustness and applicability of our findings, providing a more nuanced understanding of the molecular mechanisms involved in obesity-induced kidney disease.

In conclusion, the current study unravels the miRNA expression pattern in the OIKD mouse model in response to HFD feeding. We identify differentially expressed miRNAs and novel candidates potentially implicated in the pathogenesis of obesity-associated nephropathy. Collectively, our findings enhance comprehension of the impact of obesity and elevated lipid levels on the onset and progression of obesity-associated nephropathy at the transcriptional level, as well as offer novel biomarker candidates and therapeutic targets for managing this medical condition. Notably, we have pinpointed two endogenous control genes for qPCR analysis of miRNA expression in the OIKD mouse model, providing valuable tools for future research in the field. This study expands our understanding of molecular mechanisms and offers practical resources for ongoing investigations into obesity-associated kidney disease.

## 4. Materials and Methods

### 4.1. Animals and Experimental Protocol

The mouse experiments conducted in this study received approval from the Animal Experimentation Ethics Committee of the University of Lleida (Ethical Approval Number CEEA 05-01/18). They adhered to all applicable ethical regulations and followed the guidelines of the European Research Council for the Care and Use of Laboratory Animals. C57BL/6J mice were obtained from Charles River (Barcelona, Spain) and were housed and maintained in a barrier facility, where pathogen-free procedures were implemented in all mouse rooms. Animals were bred and maintained under controlled temperature of 22 °C in 12 h light/dark cycles with ad libitum access to food and water. After weaning at 21 days, mice were maintained on a regular mouse chow (Harlan Teklad, Madison, WI, USA) until the commencement of the experiment. At 12 weeks of age, male C57BL/6J mice were randomly divided and placed either on a standard diet (STD; *n* = 9) containing 13 kcal% fat and 20 kcal% proteins (Harlan Teklad, Madison, WI, USA) or a high-fat diet (HFD; *n* = 9) containing 60 kcal% fat, 20 kcal% carbohydrate and 20 kcal% protein (D12492, E15742-34, SSNIFF, Soest, Germany) for 10 weeks [41,83]. Body weight was measured every week throughout the whole experiment. Individual food intake was measured at three time points during the experiment. Mice were euthanized at 22 weeks of age. Briefly, mice were induced into anesthesia by continuous isoflurane employing a calibrated anesthetic delivery machine and blood was collected by cardiac puncture after a 16 h overnight fast. Afterward, the animals underwent perfusion with PBS through the left ventricle, and organs of interest were gathered for histologic examination and molecular analysis. A portion of the kidney was fixed in 4% paraformaldehyde/PBS, followed by embedding in paraffin and/or Bright Cryo-M-Bed compound (Bright Instrument Co., Huntingdon, UK) for subsequent histological and immunohistochemistry investigations. The rest of the kidney tissue was snap frozen in liquid nitrogen and kept at −80 °C until protein and RNA extractions.

### 4.2. Biochemical Analysis

Serum triglycerides, as well as total HDL and LDL cholesterol, were assessed through conventional clinical procedures employing a multichannel Hitachi Modular analyzer (Roche Diagnostics, Indianapolis, IN, USA). Serum levels of blood urea nitrogen (BUN) and glucose were determined using the commercially available colorimetric Urea-37 kit (ref.1001325, Spinreact, Barcelona, Spain) and the Glucose-TR kit (ref.1001190, Spinreact, Barcelona, Spain), respectively. Serum levels of aspartate aminotransferase (AST/GOT) and alanine aminotransferase (ALT/GPT) were determined using the commercially available AST/GOT kit (ref. 11830, BioSystems, Barcelona, Spain) and the ALT/GPT kit (ref. 11832, BioSystems, Barcelona, Spain), respectively. Serum C-pepetide levels were determined using the commercially available ELISA kits from Millipore Co., (EZRMCP2-21K, Burlington, MA, USA). Insulin sensitivity index was computed using glucose and C-peptide concentrations in the equation: HOMA-IR C-peptide = FSG (fasting serum glucose, mM) × FSP (fasting serum C-peptide, pM)/22.5 [84].

### 4.3. Histopathological Analysis

Paraffin-embedded kidney sections (5 μm) were stained by periodic acid-Schiff (PAS) technique and histologically reviewed by two independent researchers following the pre-established criteria. Kidney injury (kidney injury score) was evaluated by scoring the loss of brush border, cytoplasmic vacuolation in renal tubule epithelium, mesangial matrix expansion and thickness of the glomerular basement membrane. The scoring of 5 to 10 randomly selected, non-overlapping fields (20× original magnification) of each sample was done using a 5-point scale, as described previously [85]: 0 = no damage, 1 = 1–10% of damaged kidney cortex, 2 = 10–25% damage, 3 = 25–50% damage, 4 = 50–75% damage, 5 = more than 75% damage. The reliability of such scores for the interpretation of renal damage has already been reported [85]. 

To assess renal lipid content, Bright Cryo-M-Bed-embedded frozen kidney sections (8 μm) were fixed with 4% paraformaldehyde/PBS and stained with 0.6% Oil Red O for 10 min at RT. After rinsing with distilled water, the sections were counterstained with Hematoxylin (MAD-108.1000, Wedel, Germany) for 1 min at RT and mounted with Fluoromount-G Southern Biotech 0100-01. Stained tissue sections were examined using an Olympus BX50 microscope with an Olympus automatic camera system. The stained area of each slide was determined applying color thresholding and measuring area fractions with ImageJ software 1.53k (NIH Public Domain, RRID: SCR_003070).

### 4.4. Quantitative Real-Time PCR for mRNA Quantitation

Total RNA was extracted from kidney cortex using TRIzol reagent (MRC-TR-118, Molecular Research Center, Inc., Cincinnati, OH, USA) upon homogenization in TissueLyser (Qiagen, Hilden, Germany) (50 Hz, 30 s, 3 cycles). The reverse transcription was done utilizing the First Strand cDNA Synthesis Kit (Applied Biosystems, Madrid, Spain), following the manufacturer’s instructions. Real-time PCR, employing gene-specific TaqMan probes, was conducted on a CFX Real-Time PCR detection system (Bio-Rad Laboratories, Madrid, Spain) using TaqMan Universal PCR Master Mix, No AmpErase UNG. The procedure involved forty cycles at 95 °C for 15 s and 60 °C for 1 min. Relative mRNA levels were determined using standard formulae (ΔΔCt method), with TBP serving as an endogenous control. The results referred to a randomly selected basal sample considered as value = 1.0. Gene-specific probes used in this study were mouse MCP1 (Mm00441242_m1), TNFα (Mm00443258_m1), RANTES (Mm01302427_m1), iNOS (Mm00440502_m1), αSMA (Mm01546133_m1), Fibronectin (Mm01256734_m1), TGFβ1 (Mm00441724_m1), TBP (Mm00446971_m1) and NFκB p50 (Mm00476361_m1) (Life Technologies S.A., Madrid, Spain), and FATP2 (Mm.PT.58.12313563) and SREBP1 (Mm.PT.58.8508227) (IDT Integrated DNA Technologies, Coralville, IA, USA).

### 4.5. microRNA Sequencing and Data Analysis

miRNA sequencing (miRNA-seq) protocol including sample preparation, library preparation and sequencing, read mapping and quantification of gene expression were done in Qiagen facilities in Germany.

#### 4.5.1. Sample Preparation

RNA was isolated from 7.9–17.8 mg of kidney cortex tissue using the miRNA Rneasy Plus Universal Mini Kit (Qiagen, Hilden, Germany) according to the manufacturer’s instructions with an elution volume of 30 μL.

#### 4.5.2. Library Preparation and Sequencing

The library preparation was done using the QIAseq miRNA Library Kit (Qiagen, Hilden, Germany) (*n* = 3 for both the STD and HFD mice). A total of 500 ng total RNA underwent conversion into miRNA NGS libraries. Following adapter ligation, UMIs were incorporated during the reverse transcription phase. Subsequently, the cDNA underwent amplification through PCR (13 cycles), with the addition of indices during this process. Post-PCR, sample purification took place. The library preparation underwent quality control using capillary electrophoresis (Tape D1000). Based on insert quality and concentration measurements, the libraries were pooled in equimolar ratios. Quantification of the library pool(s) occurred through qPCR. Finally, the library pool(s) underwent sequencing on a NextSeq (Illumina Inc., San Diego, CA, USA) instrument following the manufacturer’s instructions (1 × 75, 1 × 8). The raw data were demultiplexed, and FASTQ files for each sample were generated using the bcl2fastq2 software v2.20 (Illumina Inc.).

#### 4.5.3. Read Mapping and Quantification of Gene Expression

The primary analysis was conducted utilizing CLC Genomics Server 21.0.4 and the workflow “QIAseq miRNA Quantification” within CLC Genomics Server, utilizing standard parameters for mapping reads to miRBase version 22. The processing steps included trimming common sequences, UMIs, and adapters, as well as filtering reads with lengths < 15 nt or >55 nt. Deduplication based on UMIs was performed, grouping reads that (1) started at the same position based on the end of the read to which the UMI was ligated (i.e., Read2 for paired data), (2) were from the same strand and (3) had identical UMIs. Singletons (groups that contain only one read) were merged into non-singleton groups if the singleton’s UMI could be converted to the UMI of a non-singleton group by introducing an SNP (the biggest group was chosen). Reads not mapping to miRBase, neither with perfect matches nor as isomiRs (maximum 2 mismatches and/or alternative start/end position of 2 nt), were then mapped to the mouse genome GRCm38 with ENSEMBL GRCm38 version 98 annotation, using the “RNA-Seq Analysis” workflow of CLC Genomics Server with standard parameters. Differential expression analysis utilized the ‘Empirical analysis of DGE’ algorithm in CLC Genomics Workbench 21.0.4, implementing the ‘Exact Test’ for two-group comparisons by Robinson and Smyth [86] incorporated in the EdgeR Bioconductor package [87]. For unsupervised analysis, only miRNAs with at least 10 counts summed over all samples were considered. A variance-stabilizing transformation was performed on the raw count matrix using the function vst of the R package DESeq2 version 1.28.1, and 35 genes with the highest variance across samples were selected for the hierarchical clustering.

### 4.6. Quantitative Real-Time PCR for miRNA Validation

To confirm miRNA-seq results, we used quantitative real-time PCR (qPCR) to measure the expression of selected miRNAs in the mouse kidney. Reverse transcription reaction (cDNA synthesis) was performed using the miRCURY LNA RT kit (Qiagen), and real-time PCR with specific miRCURY LNA miRNA PCR Assays (Qiagen) was carried out with a QuantStudio 7 Real-Time PCR detection system (Applied Biosystems) using miRCURY LNA SYBR Green PCR kit (Qiagen). Total reaction volume for qPCR reaction was 10 μL. Forty cycles at 95 °C for 10 s and 56 °C for 60 s min were performed, followed by melting curve analysis. Relative miRNA levels were calculated by standard formulae (ΔΔCt method) using miR-30a-3p and miR-30b-5p as endogenous controls. Only candidates with high expression levels were subjected to stability analysis (expressed in 100% of samples and expressed in the first quartile (Q1)). Endogenous controls were selected combining statistical tools like NormFinder, geNorm, Best-Keeper and Delta Cq/Ct algorithms that are integrated within the RefFinder software [88]. Specific miRCURY LNA miRNA PCR Assays (Qiagen) were as follows: miR-5099 (YP02107868), miR-223-3p (YP00205986), miR-551b-3p (YP00204067), miR-21a-3p (YP00205400), miR-146a-3p (YP02115408), miR-30a-3p (YP00204457) and miR-30b-5p (YP00204765).

### 4.7. Western Blot Analysis

Proteins were extracted from the kidney cortex by homogenization in Triton-soluble buffer (50 mM Tris (pH 7.5), 150 mM NaCl, 1% Triton X-100, 0.5 μM EDTA, 1 mM Na_3_VO_4_, 2 mM PMSF and protease inhibitor cocktail) using TissueLyser (Qiagen, Hilden, Germany) (50 Hz, 30 s, 3 cycles). Tissue lysates were subsequently rotated at 4 °C for 1 h and centrifuged at 13.000 rpm for 15 min at 4 °C. Supernatant was saved at −80 °C. Protein concentrations were determined using a DC protein assay kit (Bio-Rad). A total of 20 μg of proteins were treated for 30 min at 50 °C in a loading buffer containing 2% SDS and 5% β-mercaptoethanol and electrophoresed, as explained previously [38]. Briefly, the samples were subjected to electrophoresis on 8% or 10% SDS-PAGE gels, as appropriate, and transferred to PVDF membrane (pore size 0.45 μm, Immobilon-P, Millipore). Membranes were blocked for 1 h with 5% skim milk in Tris-buffer saline solution containing 0.1% Tween-20 (TBST) and subsequently probed with primary antibody against phospho-specific Akt (Ser 473) (#4060S, Cell Signaling; 1/2000), phospho-specific Erk1/2 (Thr 202/Tyr 204) (#675502, BioLegend, San Diego, CA, USA; 1/5000), total Erk1/2 (#686902, BioLegend; 1/1000), phospho-specific p-38 (Tyr 182) (E-1, sc-166182, Santa Cruz, Santa Cruz, CA, USA; 1/1000), Ik-Bα (sc-371, Santa Cruz; 1/1000), and GAPDH (#919501, BioLegend), overnight at 4 °C. Horseradish peroxidase-conjugated secondary antibodies (anti-mouse, #115-035-003, Jackson Immunoresearch, West Grove, PA, USA; anti-rabbit, #7074, Cell Signaling; anti-rat, #112-035-003, Jackson Immunoresearch) were used at 1/10.000 for 1 h at RT. Visualization of the immunoreaction was performed using chemiluminescent kits EZ ECL (Biological Industries, Beit HaEmek, Israel) or ECL Advanced (Amersham Biosciences, Amersham, UK). Images were digitally acquired by ChemiDoc™ MP Imaging System (Bio-Rad).

### 4.8. Statistical Analysis and Bioinformatics

Statistical analysis was conducted using GraphPad Prism 9 software (GrahPad Software, San Diego, CA, USA) and R Statistical Software (v4.3.2; R Core Team 2023). Data are expressed as mean ± standard error of the mean (SEM) of each group. The normality of the distribution was tested using the Shapiro–Wilk test and the homogeneity of variance using the Levene test. If the normality assumption was met, a comparison between the two groups was conducted using the Student *t*-test. However, in cases where samples did not adhere to a Gaussian distribution, means were compared using the Mann–Whitney statistical test. Depending on the homogeneity of variance, means were compared using the Student *t*-test with or without Welch correction. *p* < 0.05 was considered statistically significant. Bioinformatic prediction analysis was assessed using the web-based computational tool DIANA-miRPath v4.0 [37]. DIANA-miRPath v4.0 combines information on predicted miRNA:gene interactions from Targetscan v8.0 with the Kyoto Encyclopedia of Genes and Genomes (KEGG) database and Gene Ontology (GO) annotations (for biological processes). 

## Figures and Tables

**Figure 1 nutrients-16-00691-f001:**
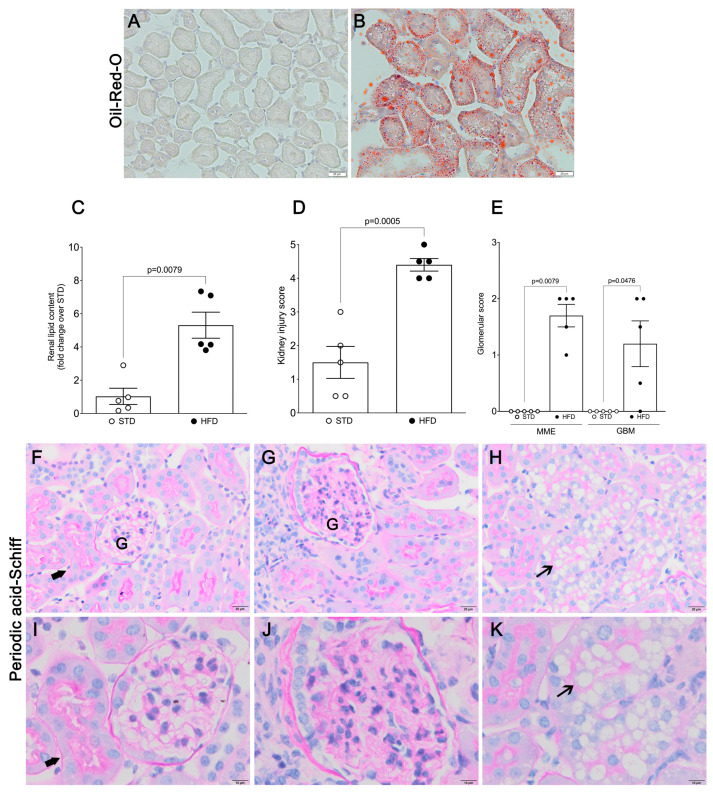
Histological assessment of tubular and glomerular changes upon HFD feeding. Representative photomicrographs of Oil Red O (**A**,**B**) and periodic acid-Schiff staining (PAS) (**F**–**K**) of frozen (**A**,**B**) and paraffin-embedded (**F**–**K**) kidney sections from mice fed an STD (**A**,**F**,**I**) and an HFD (**B**,**G**,**H**,**J**,**K**). Thick arrows, healthy renal tubule; thin arrows, vacuolization of tubular cells; (**A**,**B**,**F**–**H**) scale bar represents 20 μm, (**I**–**K**) scale bar represents 10 μm. (**C**) Quantification of renal lipid content (Oil Red O). (**D**) Kidney injury score. (**E**) Histopathological assessment of glomerular parameters. Data present the mean ± SEM of 5 mice/group. Three tissue sections per animal were analyzed. MME, mesangial matrix expansion; GBM, glomerular basement membrane thickness; G: Glomerulus.

**Figure 2 nutrients-16-00691-f002:**
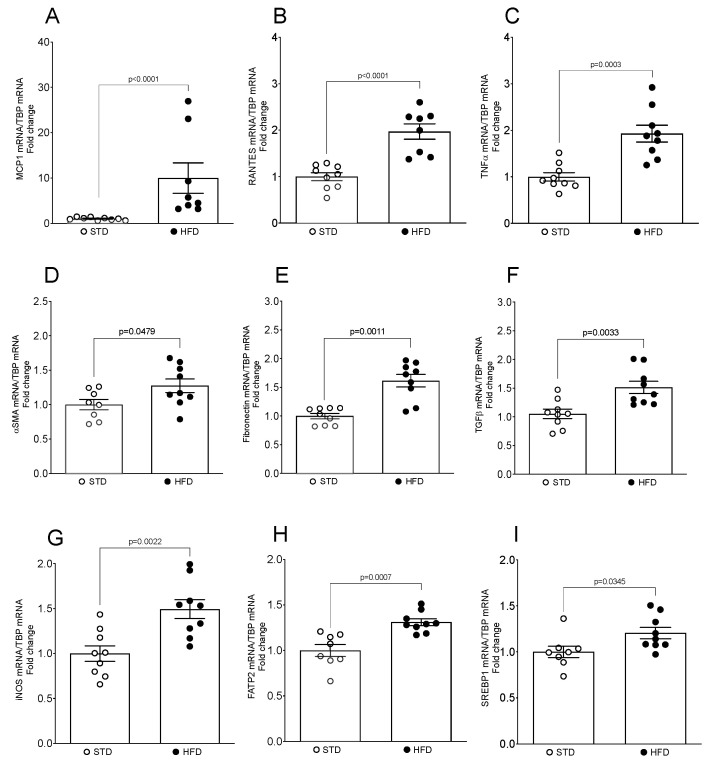
Expression levels of obesity-related genes and inflammatory and profibrotic markers in the mouse kidney upon HFD feeding. (**A**–**I**) Total mRNA was isolated from kidneys and mRNA levels for specific genes were assessed by real-time qPCR. The relative mRNA levels were calculated and expressed as fold change over STD (value = 1.0) after normalizing for TBP. Data are presented as mean ± SEM (*n* = 8–9 mice/group).

**Figure 3 nutrients-16-00691-f003:**
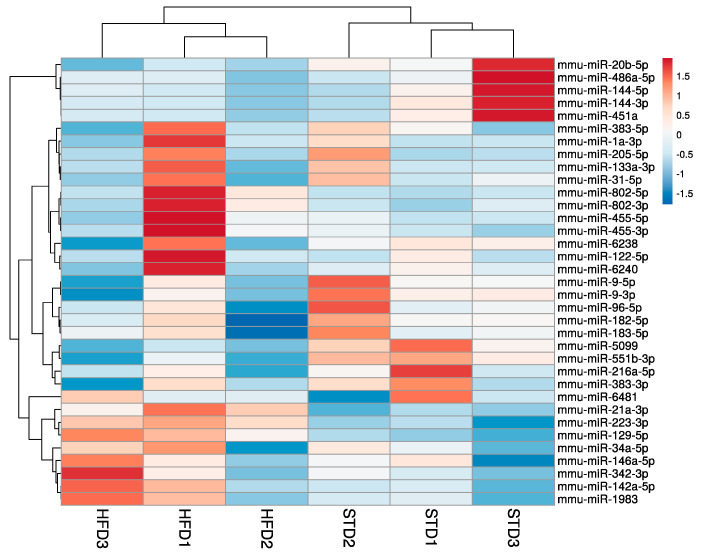
Hierarchical clustering of most variable miRNAs in a mouse model of OIKD. The heatmap was generated by the unsupervised hierarchical clustering of the miRNA profiles of 3 STD- and 3 HFD-fed mice. A variance-stabilized transformation was carried out on the raw count matrix, and hierarchical clustering was performed on the top 35 genes with the highest variance across samples. Each row in the matrix corresponds to an individual gene, while each column corresponds to a distinct sample. Rows are centered; unit variance scaling is applied to rows. Both rows and columns are clustered using correlation distance and average linkage.

**Figure 4 nutrients-16-00691-f004:**
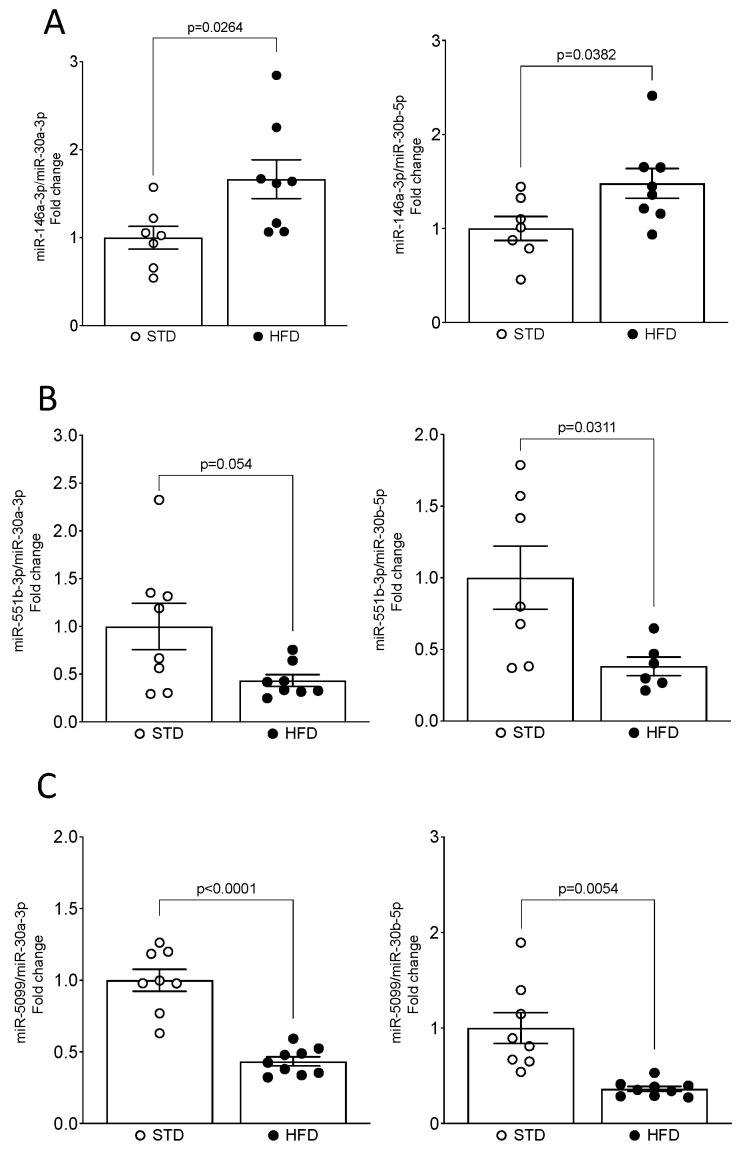
qPCR validation confirmed modulation of miR-5099, miR-551b-3p and miR-146a-3p in the mouse kidney upon HFD feeding. (**A**–**C**) miRNAs levels were determined by real-time qPCR and expressed as fold change over STD (value = 1.0) after normalizing for both miR-30a-3p and miR-30b-5p, as endogenous controls. Data are presented as mean ± SEM (*n* = 7–9 mice/group).

**Figure 5 nutrients-16-00691-f005:**
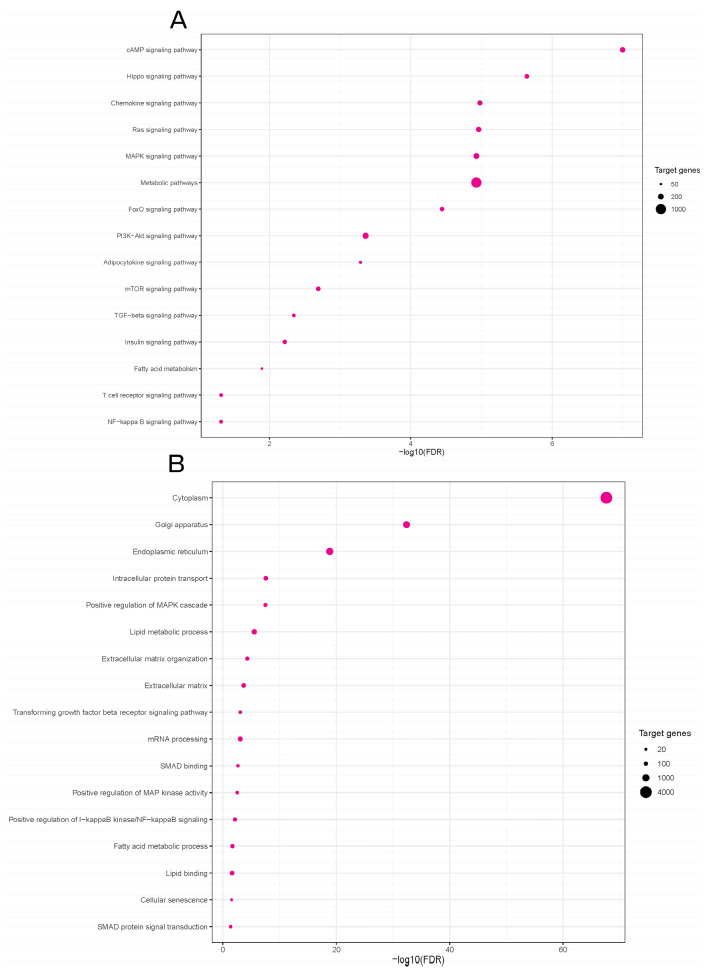
Kyoto Encyclopedia of Genes and Genomes (KEGG) and Gene Ontology (GO) analyses of the predicted miRNA target genes in a mouse model of OIKD. (**A**) KEGG biological pathway analysis (selected). (**B**) GO functional enrichment analysis for biological processes (selected). The *p*-value represents the significance of the biological process and molecular pathway. The size of the points represents the number of genes involved in the process or pathway. The false discovery rate (FDR)-adjusted *p*-value cutoff was 0.05.

**Figure 6 nutrients-16-00691-f006:**
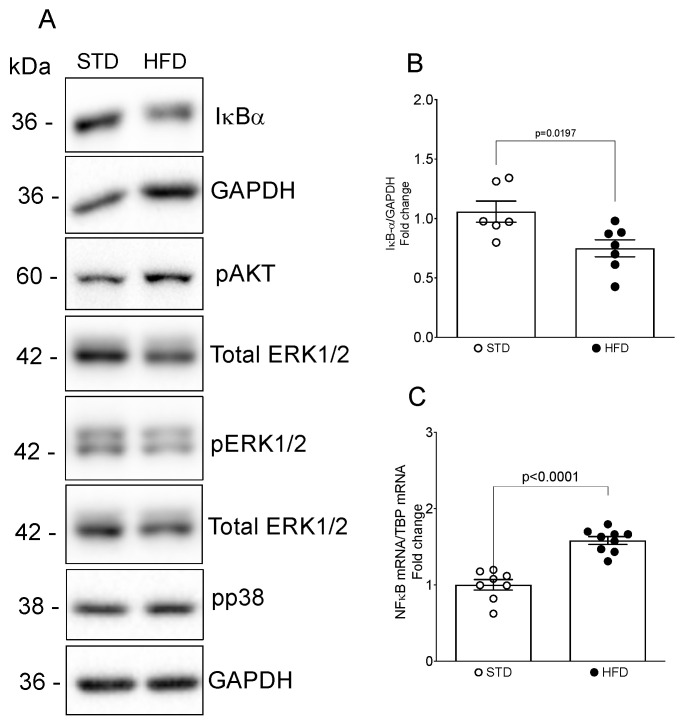
Evaluation of NFκB, PI3K-Akt and MAPK pathway activation in the mouse kidney upon HFD feeding. Whole kidney lysates underwent protein analysis and were immunoblotted with antibodies against pAkt, pERK1/2, p-p38, Iκ-Bα, total ERK1/2 and GAPDH. (**A**) Representative Western blots. (**B**) Quantitative densitometric analysis. Data were normalized to GAPDH and presented as mean ± SEM (*n* = 6–7 mice/group) (fold change over STD). (**C**) mRNAs levels for NFκB p50 were determined by real-time qPCR and expressed as fold change over STD (value = 1.0) after normalizing for TBP. Data are presented as mean ± SEM (*n* = 8–9 mice/group).

**Table 1 nutrients-16-00691-t001:** Biochemical and metabolic parameters of STD vs. HFD group.

Parameters	STD	HFD	*p*
Body weight (g)	32.78 ± 0.662	41.78 ± 1.164	<0.0001
Food intake (g/mouse/24 h)	3.69 ± 0.193	2.90 ± 0.050	<0.0001
Total cholesterol (mg/dL)	100.1 ± 4.478	148.4 ± 6.679	<0.0001
LDL cholesterol (mg/dL)	10.65 ± 1.369	16.75 ± 2.238	0.0356
HDL cholesterol (mg/dL)	78.11 ± 3.728	114.2 ± 4.983	<0.0001
Triglycerides (mg/dL)	64.13 ± 3.404	94.78 ± 10.80	0.0008
Glycemia (mg/dL)	148.2 ± 9.843	197.4 ± 15.98	0.0229
BUN (mg/dL)	26.33 ± 1.373	21.09 ± 0.7058	0.0037
AST/GOT (IU/L)	8.54 ± 1.268	17.65 ± 1.877	0.0014
ALT/GPT (IU/L)	1.60 ± 0.549	5.50 ± 1.654	0.0508
C-peptide (pM)	33.72 ± 3.845	75.04 ± 28.10	0.0288
HOMA-IR index	12.92 ± 1.931	39.48 ± 17.94	0.0206

BUN, blood urea nitrogen; AST/GOT, aspartate aminotransaminase; ALT/GPT, alanine aminotransaminase.

**Table 2 nutrients-16-00691-t002:** List of miRNAs showing q-value < 0.05 in the comparative analysis of STD vs. HFD groups.

miRNA	Identifier	Fold Change	FDR *p*-Value (q-Value)
mmu-miR-5099	URS000056F567_10090	−2.624887175	6.4841 × 10^−14^
mmu-miR-223-3p	URS00000B7E30_10090	1.481082774	4.41731 × 10^−5^
mmu-miR-551b-3p	URS000008C563_10090	−1.813165613	0.000856739
mmu-miR-21a-3p	URS000015930E_10090	1.745000997	0.005360045
mmu-miR-146a-3p	URS0000121576_10090	2.179056951	0.005360045
mmu-miR-129-5p	URS00004E1410_10090	2.420949709	0.012797245
mmu-miR-142a-5p	URS00001E0AEA_10090	1.285697411	0.02634235
mmu-miR-144-3p	URS000037C5A8_10090	−3.132081075	0.028191062
mmu-miR-802-5p	URS00005B2C05_10090	1.682896286	0.034999635

## Data Availability

Data is contained within the article and Appendix A.

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
