# Peer review of "microRNA Expression Profile in Obesity-Induced Kidney Disease Driven by High-Fat Diet in Mice"

_nutrients, 2024, doi:10.3390/nu16050691_

Round 1

Reviewer 1 Report

Comments and Suggestions for Authors

In this study the authors demonstrate that a high fat diet induces clear features of injury in the kidney. They subsequently performed RNA-seq and identified miRNAs associated with this obesity-induced kidney injury. Finally, potential pathways targeted by these miRNAs are identified and it is shown that amongst others the nf-kb pathway is altered in HFD.

This is an interesting and well written study and, in my opinion, obesity-induced kidney damage has gained too little attention, thus novel studies investigating its (potential) pathophysiology are important.

I have some minor comments:

It would be beneficial to better define kidney damage in the cortex (that is used for the injury score in figure 1D); what are the exact features that have been checked? (tubular necrosis?  Or is it the features mentioned for figures 1GJ: larger glomeruli, mesangial matrix expansion (MME), thickening of the glomerular basement membrane?

Figure 1A is only described after the other parts of the figure, I personally would prefer to move fig1A to the end of the figure then.

Figure 2: Although a bit semantics, in my view a 0.97-fold change means that the expression in the new situation is 97% of the original situation (so a decrease of 3%). However, I assume that for e.g. RANTES the new expression levels is 97% higher (=1.97 fold change). I would advise to change the fold-change values accordingly.

Figure 4: even though the other 2 miRNAs were not validated by qPCR, it would be beneficial to show their qPCR data as well.

Figure 6: I think the legend for figure 6C is incorrect? It mentions miRNAs, but it displays nf-kb? (and also says mRNA?) But this one is not found back in figure 6a? it does show GAPDH twice so should the second blot be NFkB p50?

I assume HFD also induces high blood pressure, this may also affect kidney phenotype and therefore miRNAs. This may be discussed as another secondary contributor to the phenotype/miRNAs.

Reviewer 2 Report

Comments and Suggestions for Authors

Ebitja et al present an original research article entitled "microRNA Expression Profile in Obesity-Induced Kidney Dis-2 ease Driven by High-Fat Diet in Mice".

In this work they used a model of obesity-induced kidney disease in C57BL/6J mice in whic they perfomed next generation sequencing (NGS) analysis with a focus on microRNA. They found that 9 miRNAs were differentially expressed in the kidney including miR-5099, miR-551b-3p, miR-223-3p, miR-146a-3p and miR-21a-3p

This article is important as a lot of miRNAs were involved in obesity in the litterature but few RNA-Seq systematic work was used. the model is

Major revision:: Some miRNAs listed ahve been incriminated in obesity-related diseases such as diabetes, Chronic kidney disease, and cardiovascular disorders and that should be discussed a bit more. For example miR-223 has been implicated in CKD. See for example review in PMID: 30891535

also the same miRNA has been implicated in NFKB pathway, and that is intersting since the authors focus on this pathway heavily. See PMID: 29778662

what were the criteria used to check RNA integrity?

How did the authors standardize the parts of the kidney that were used for experiments? Whole kidney, cortex, left part, right part etc?
